# Earthquake Resilient near Zero Energy Buildings: Attributes and Perspectives

**Anthimos Anastasiadis [1] and Marius Mosoarca [2,*]**

1   ASAnastasiadis & Associates, T. Papageorgiou 10, 54631 Thessaloniki, Greece; anastasiadis@asacon.eu
2   Department of Architecture, Politehnica University of Timisoara, Traian Lalescu 2A,
    300223 Timisoara, Romania
*   Correspondence: marius.mosoarca@upt.ro

**Abstract:** The climate crisis, the need for a circular economy, and the large financial losses after earthquakes have promoted the concept of the sustainable and resilient design of societies, and more specifically, of lifelines and building environments. Focused on building facilities, it is imperative to prescribe, within the aforementioned framework, the components that characterize earthquake resilient near zero energy buildings (ERnZEBs). Through a conceptual analysis, the goal is to discuss the attributes and perspectives of ERnZEBs within the framework of the view of a designer engaged in practice. This fact introduces an additional factor recognizing that not all projects have the same technical and financial values; the difference in budget, the type of owner, and the investment (private or public, company or private person) play important roles in creating an ERnZE building. In this direction, this paper reviews the basic principles of ERnZEBs, providing a combination of pragmatic considerations while also exploiting the state of the art and practice of current engineering knowledge.

**Keywords:** earthquake sustainability; earthquake resilience; holistic design; construction technologies

## 1. Introduction

Starting from the principle of Vitruvius [1], that all buildings must have three characteristics, namely *Firmitas* (Strength), *Utilitas* (Utility), and *Venustas* (Beauty), and moving on to the current strong urbanization that needs sustainable and resilient societies against multiple hazards (earthquakes, floods, tsunamis, fires, and explosions) [2–7], it is of paramount importance to define the attributes and perspectives of the basic components of communities and societies, which include new and existing building facilities. Definitely, buildings are a sub-system as compared with the whole system, but they make a great contribution to the built environment.

Focused on earthquakes, major studies were performed and published that were associated with building resilience [8–11], providing framework methodologies and tools generally based on probability. To this end, one can mention that sellers like real estate developers, insurers, reinsurers, and bankers, as well as asset managers, think about and take decisions in a probabilistic way. This is due to the fact that probability is never unfair and shares responsibility and liability as well. Instead, buyers of buildings are characterized by a deterministic way of thinking. They want to buy an asset that will be durable and resistant for a long time, with minimum maintenance, and without structural damage, building downtime, or financial losses in its life cycle. This is true for someone who buys, although it is not true for technical and financial stakeholders. Therefore, there is a gap between them. The building's environment and seismic action are inherently variable and uncertain. Although it should be recognized that seismic risk is strongly influenced by political decisions, it is more specifically a tradeoff between risks and costs for a given known hazard. The question that arises is: how much does a typical homeowner know

about the acceptable risk? Moreover, what is the financial capacity of a typical homeowner to take a risk? Unbeknownst to the homeowners, this risk has been taken by the design code without their even knowing it.

Coming back to the Vitruvius attributes and translating them into the current construction practice, a building must be aesthetic, structurally durable against environmental actions, founded on stable soil with a structurally sufficient bearing capacity against static (i.e., dead and live actions) and dynamic loads (i.e., wind and seismic actions), water resistant, thermally and acoustically efficient, as well as fireproof. A building should not only be mechanically resistant from geotechnical and structural points of view; it is not sufficient. The aforementioned attributes define only an Earthquake-Resistant Energy Efficient Building.

Nowadays, this is not sufficient. The climate crisis, the need for a circular economy, and the large financial losses after earthquakes [12,13], promoted a twofold target: (i) a shift from fail-safe to safe-fail [3], or specifically, a design shift from collapse prevention and life safety to a resiliency towards full pre-earthquake functionality [14,15] and (ii) the development of buildings that are characterized by a very high-energy performance during operation and where most of the energy required is provided by energy from renewable sources (typically solar thermal and photovoltaic (PV) systems) [16] (or the new, very optimistic proposal of zero emission buildings [17]). With regard to the first target, beyond the stiffness, strength, and ductility, in addition, the adaptability and reparability capacities within a tolerable timeframe after an earthquake were considered. Related to the second target, buildings must have the necessary system supplies along with efficient insulating composite systems that limit the HVAC's (heating, ventilation, air conditioning) consumption.

According to the above-mentioned issues, such buildings could be described as Earthquake Resilient near Zero Energy Buildings, ERnZEBs. Consequently, this prototype building has the following attributes: the ability to be durable against environmental actions; to avoid foundation soil failure; to prevent collapse and to protect life safety; to develop, after an earthquake, repairable damage to structural and non-structural elements; to recover and restore its occupancy and functionality after an acceptable time (after an earthquake); to be allocated very high-performance systems, reducing energy consumption and $CO_2$ emissions; to be waterproof, fire resistant, and acoustically efficient. The first five characteristics are connected with geotechnical and structural design, while the remainders are associated with building physics. Certainly, architectural functional adaptability is a prerequisite for such buildings. ERnZEBs go beyond the current trend of seismic resilience and energy efficiency. Equally, it must be taken into account that there is a need for durability, water, and fire protection, as well as acoustic comfort. In any case, all the elements, connections, and systems that offer the benefit of near zero energy buildings must be earthquake resilient. Finally, ERnZEBs must have similar attributes for new and existing buildings, taking into account the construction period and the desired level of improvement. Overall, the two pivotal pillars of an ERnZEB are sustainability and resiliency; a strong interdependency between them is required.

Through a conceptual analysis, the goal is to discuss the structural attributes and perspectives of ERnZEBs, within the framework of a designer's view who is engaged in practice. This fact introduces an additional parameter, namely, that all the projects are not the same; the difference of budget, the type of owner, and the investment type (private or public, company or private person) play important roles in creating an ERnZE building. Furthermore, there should be consideration of the local insights into the task, tailored to a building facility, not only global insights at a policy level. Therefore, this paper reviews the principles of an ERnZEB combining a pragmatic approach while also exploiting the current state of the art and practice of structural engineering knowledge. It is focused on the low- and mid-rise buildings having different uses, namely, residential, health care, hospitality facilities, and commercial. The structural materials mainly considered are reinforced concrete and structural steel. This work is based on the concept that, even though they behave differently (due to the structural material), buildings are fundamentally

designed according to capacity design, and thus have the same plastic mechanisms (global plastic mechanism where the plastic hinges are developed only into the beams and not to the columns, avoiding soft-story brittle mechanisms).

## 2. Earthquake Sustainability of ERnZEBs

The climate crisis and the resulting environmental changes strongly promote sustainable development. According to the UN's Brundtland Commission [18], economic growth for a sustainable society must meet the needs of the present without compromising the ability of future generations to meet their own needs. However, as reported by the World Green Building Council [19], the building sector (production of building materials, transportation, construction, and demolition) is responsible for 39% of global carbon emissions. Focused on the consequences of strong earthquakes, and more advanced than severe damage, are the collapsed buildings and their demolition wastes, a serious problem that perturbs sustainability. Moreover, the collapsed buildings should be replaced with new ones. As an example, approximately 70% of the district of Christchurch, after the 2011 earthquake, was demolished (namely, over 60% of the reinforced concrete building with three stories and more, around 1000 commercial properties, and 10.000–15.000 residential properties) [20,21]. The recent Kahramanmaras earthquake in Turkey in 2023, which affected nearly 16 million people, resulted in approximately 280.000 buildings collapsing or being severely damaged; hence, it was the second most severe case of post-earthquake demolition [22]. All of the above-mentioned real facts violate the three basic principles of sustainability: *Reduce*, *Reuse*, and *Recycle*. It is, therefore, clearly stated that an ERnZEB must respect the rule of Rs. It should be valid for both new and existing building structures. However, it should not be forgotten that, in earthquake-prone countries, sustainability continues through the earthquake. This means that every energy and environmental design should be supported by a resilient structural design. Otherwise, there is a loss of investment; see Figure 1.

Evidently, sustainable construction is limited by the use of energy minimization, raw materials, gas emissions, and waste generation and management. Earthquake sustainability, and in a more general sense ERnZEBs, goes beyond this, requiring the following: the choice of highly recycled materials, durability, efficient use of structural, recycled materials exploiting their potential capacity, awareness of geological and local geotechnical soil foundation conditions, choice of efficient foundation systems, selection of structural systems that offer mitigation of displacements, equilibrated structural conformation layouts, components combining different functions (i.e., structural and energetic, energetic and mechanical, etc.), selection of details that minimize maintenance and repairs, easy erection and demolition, structural repairability following an earthquake, detailing by considering the disassembly and reuse of materials or elements. Certainly, apart from the others, we do not forget that everything starts with the architecture synthesis. Therefore, the functional adaptability of a building facility is of primary importance; practically all the aforementioned attributes will be maximized if the architectural layout permits the change of use and the possibility of renovation.

The drivers of sustainability, specifically, the strategy of reduce, reuse, and recycle, (reduce the production of construction materials, reduce the waste, reuse all the building components and elements, recycle everything from a building facility), within the framework of seismic action, are discussed below and further explained according to ERnZEB perspectives.

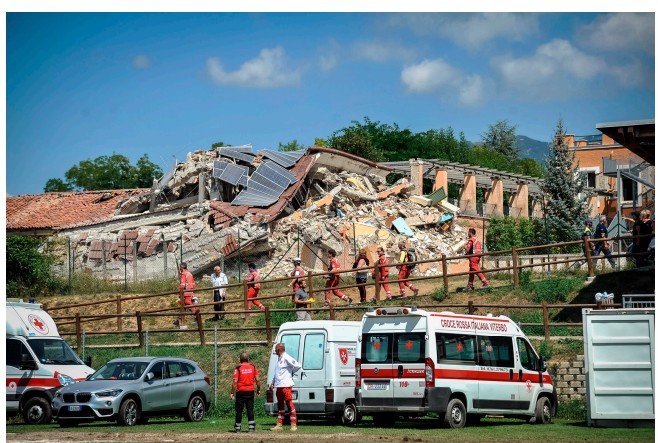

(**a**)

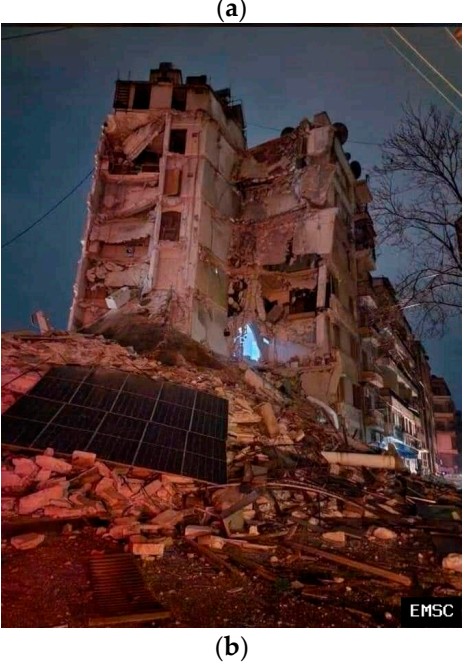

(**b**)

**Figure 1.** (**a**) L' Aquila earthquake, Italy, 2009, and (**b**) Kahramanmaras, Turkey, 2023. (Photos from European-Mediterranean Seismological Centre).

### 2.1. Reduce

Addressing the issue of structural material reduction, for the construction of an earthquake-resistant building (and, in a more advanced version, an earthquake resilient building structure), one can easily observe a controversial relationship between the first sustainable rule and the construction of a new building after a catastrophic earthquake (or the obsolescence of an existing building after a catastrophic seismic action). Systematically, after each strong earthquake, the codified seismic forces increase; the dimensions of the structural elements increase, consequently the consumption of building materials also goes up. Such an example is the case of Romania. For the capital city of Bucharest, from 1963 until now, the seismic design acceleration has increased 12 times, while for other county cities, this increase achieved 4 to 10 times [23]. It is a typical case for every earthquake prone country (Greece, Italy, Turkey, Japan, USA, New Zealand).

Another debatable issue is the well-established use of the behavior factor q (or response factor R according to US practice). For instance, in agreement with the forced-based design in EN 1998:1-1 [24], for the low ductility level, the q-factor is equal to 1. This fact leads to a structure that is designed elastically (of course without ductile detailing, as prescribed in the respective code), and it is not permitted to be applied in high seismicity areas. For a high ductility level, corresponding to a dual frame with a regular layout and cross section,

the q-factor attains a value of 4.95. This means that the elastic design force is reduced approximately five times (strictly respecting ductile detailing). Evidently, in a disastrous earthquake, life safety would be protected; however, with so much damage, the building must be demolished. This was demonstrated by the New Zealand earthquakes [15]. Overall, the goals of sustainability are at the opposite end. A concept to design with a q-factor equal to one or to one and a half, always respecting the ductile detailing, will lead to a sustainable ERnZEB. According to unpublished studies of the authors, the average cost increase will be of the order of 20–25% (for a spectral acceleration of 0.36 g). Overall, an earthquake sustainable proposal occurs when we use a force based design to perform an elastic analysis and design with a q equal to one, or alternatively one and a half, and further on to respect the material, section, and member ductility and detailing. It is a pragmatic way of designing building until a resilient earthquake design, based on accepted target performance levels of drift, plastic rotation, crack width, accumulated deformation, and residual deformation, is developed.

Looking from a general perspective, the solution to this problem revolves around two axes. The first one is to give special attention at the phase of preliminary design, looking for an efficient conformation (suitable systems of foundation, positioning of structural walls at all principal directions of action, balanced layouts against torsion, etc.). A proper structural conformation saves lives, minimizes structural and non-structural damage, and protects properties; therefore, it reduces demolition cases, material waste, and building materials used for repair or strengthening. In many cases, this cannot be applied as desired by the structural engineer; this is due to architectural constraints. This last statement would be avoided if the architects would take into consideration the basic principles of seismic design and seismic urban planning in the same manner as sustainability.

The second axis moves toward the application of elements that change the dynamic characteristics of the building facility (i.e., base isolation) and/or control the behavior with passive damping (i.e., viscous, friction dampers, etc.), semi-active or active systems [25]. An illustrative example of the base isolation approach, along with an efficient energy design, was applied for the reconstruction after the L' Aquila earthquake, 2009, Italy, [26].

*2.2. Reuse*

This second principle of sustainability is twofold: (i) it is connected to a greater extent with existing building facilities, and (ii) the same ones, although after an earthquake. The concept of reuse must be related with another rule of 3 Rs, namely, with the Rehabilitation, Restoration, and Renovation.

From a structural point of view, the task is to retain the load-resisting system, LRS. If we have an existing building, the target is to strengthen the existing one by using the concept adapted from UNISDR to "*build it back better*" [27]. In the event that we have a building under construction, the issue is to employ structural prefabricated members (from steel, reinforced concrete, or timber) and, under certain conditions, also reuse the existing foundations (i.e., introduce micropiles, change a pad foundation system to a raft foundation, etc.).

From another perspective, designers for new buildings should already be thinking about the deconstruction of a structure. In an undamaged condition, it is relatively easy. In an earthquake-resistant environment, it is difficult to think about it. However, if the capacity design is used according to structural hierarchies, then the designer would save some members without damage, where they would be recovered and reused. Practically, this is difficult to achieve in the hard conditions after a catastrophic earthquake; as a consequence, the seismic design should be focused to generate structures that will respond elastically. To this end, one can mention that elastic stiffness is a structural property of a mechanical system that is reclaimed. The ductility is used once, and after a severe seismic action is "consumed", producing damage; therefore, it is not reversible, and as a matter of fact, it is not a structural attribute that leads a member or a structure to be reused. Speaking

in a figurative sense, ductility is not "materially sustainable", although it must be recalled that, due to its force, redistribution and deformation capacity lead to lives being saved.

### 2.3. Recycle

To recycle is not to reuse. The waste materials are converted into new ones and not transformed. Thus, the recycling process is strongly related to building materials. At the same time, the topsoil from building excavations will also be put into service for landscaping works.

For earthquake-resistant purposes, steel structures represent a viable solution due to their complete recyclability, reduced weight, strength and ductility, architectural flexibility, dry construction, capacity of dissembling and reuse. It was the building material of choice for the reconstruction of Christchurch, following the 2010–2011 series of earthquakes that completely closed the operations in the Central Business District [28]; see Figure 2. Additionally, steel structures after major worldwide earthquakes behaved excellently, presenting only local failures and not global building collapses [29].

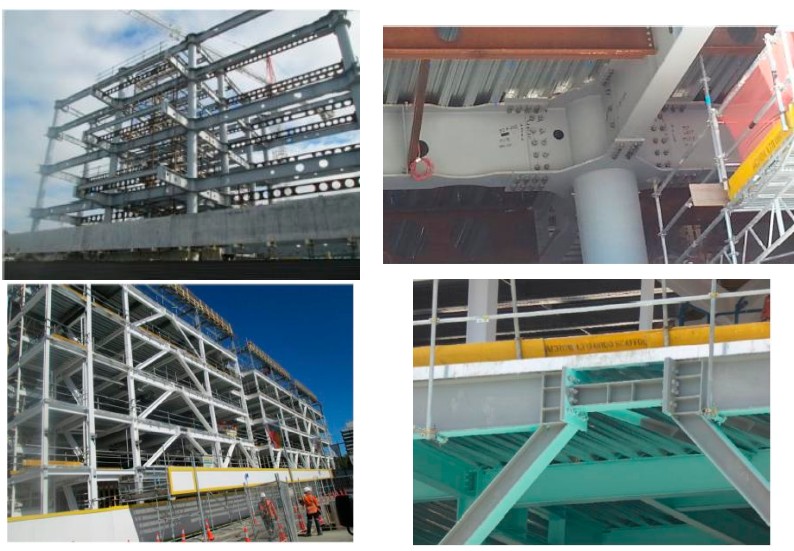

**Figure 2.** Reconstruction of Christchurch CBD using steel structural solutions [28].

A promising solution in the near future will be the construction of multi-story timber buildings, which are a recycle and sustainable solution for both the environment and human wellbeing. Structurally, it is similar to structural steel; however, it is a brittle material. Due to the fact that tall timber buildings have not been tested in strong earthquakes, investors and private owners are conservative about investing in or buying such buildings. Nevertheless, great research efforts are performed in order to better understand the cyclic behavior and, further on, to produce reliable codes that will open the door for the construction of multi-story building facilities [30–32]. Currently, cross laminate timber panels are used for both the seismic and energetic improvement of existing reinforced concrete buildings [33] and can also be applied to steel buildings.

### 3. Resiliency of ERnZEBs

Recent devastating worldwide earthquakes reveal that, not only life safety, but also financial losses and building downtime are of great importance [4,34,35]. New well-designed buildings, generally constructed after the 1990s and based on proper foundation systems, even in the case of difficult soil foundation conditions, behaved properly and demonstrated the prevention of collapse and, as a matter of fact, life safety protection. Nevertheless, the non-structural elements (i.e., partition walls, facades, ceilings, HVAC systems, etc.) were severely damaged, disrupting the building's operation and, in many cases, provoking damage to the nearby properties; see Figure 3. As was expected, regarding

the old buildings constructed in the 1950s, 1960s, 1970s, failures from severe damage to collapse were observed. Thus, for the first one, the target is to avoid non-structural damage, while for the second, the target is to "*to build back better*".

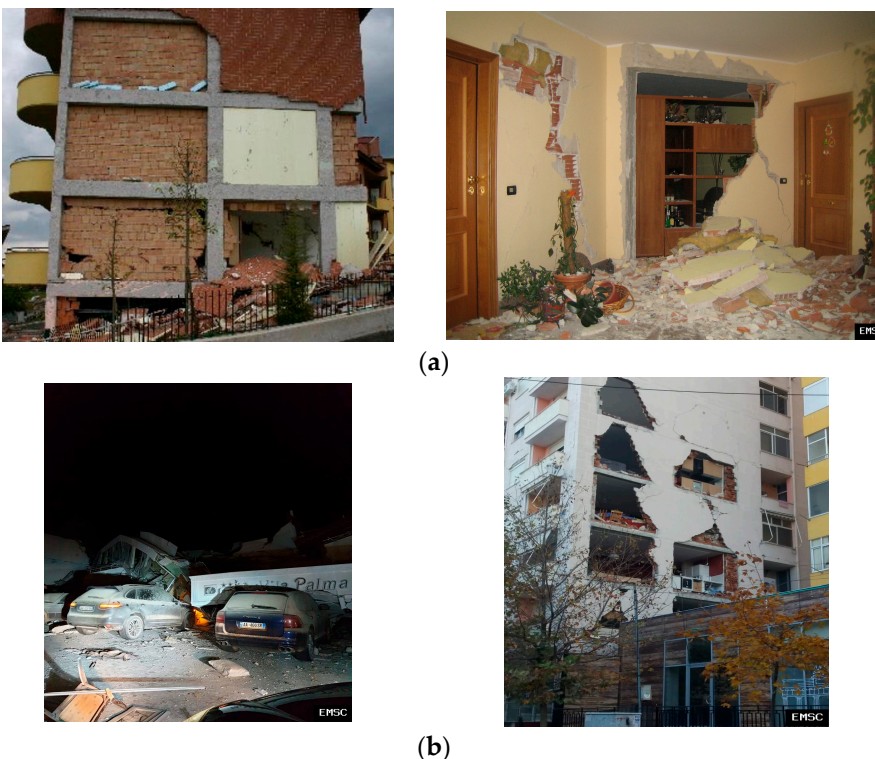

(**a**)

(**b**)

**Figure 3.** (**a**) Infill masonry failure in the L' Aquila earthquake, Italy, 2009, (**b**) infill masonry failure in the Durres earthquake, Albania, 2019.

Facing the above-mentioned issues, M. Bruneau et al. first introduced the concept of the seismic resilience of communities, putting it under discussion and proposing a framework for the "*Reduced failure probabilities*", "*Reduced consequences from failures*", and "*Reduced time to recovery*" [36–38].

Every building facility is a subsystem of a city (and more generally, a unit component of a community or a society) connected to and influenced by the performance of lifelines [6,39]. Focused on buildings, and learning from worldwide structural failures, it is imperative to start changing the position of design, moving from the classical ductility concept (which generates damage), to the concept of avoiding damage through more rigid structures (which generates repairable structures after a strong earthquake) [5–15]. The target is focused on reoccupancy, functional recovery time, and financial loss minimization [4–6]. It is a step forward from earthquake-performance-based design to resilient-earthquake-based design. In this direction, the EERI (Earthquake Engineering Research Institute) provided a definition suitable for both buildings and lifeline infrastructure, as follows: "*Functional recovery is a post-earthquake state in which capacity is sufficiently maintained or restored to support pre-earthquake functionality*" [4].

In a more general way of thinking, M. Bruneau and A. Reinhorn introduce the resilient concept in the field of earthquake design, providing the following statement: "*The seismic resilience of a system can be achieved by reducing its probability of failure during an earthquake, as well as reducing the consequences from such failures and the time to recovery*" [37]. Continuing, they provide the four attributes of a resilient system, namely: Robustness, Redundancy, Resourcefulness, and Rapidity. Each of them has four dimensions, connected with engineering, managerial–organizational, social, and financial issues [36,38].

The pivotal pillars of resiliency within the stage of seismic action are discussed below and further explained, concentrating on ERnZEB perspectives.

### 3.1. Robustness

The robustness is associated with the buildings' structural capacity to avoid damage, to continually operate after an earthquake, to not disturb the operation of the users, of the city, or of society, and, further along, to avoid financial losses. Nevertheless, the basic premise is that the utilities, water, and electric supply, sewerage (locally at the building and globally at the city level), and elevators will remain undamaged after a strong seismic action. If the building is fully operational (structurally and with all the utilities in service), then it will not be directly influenced by a probable collapse of some infrastructure components (i.e., damage to roads and bridges). Indirectly, the collapse of lifelines indicates the failure of society's resilience. There is no possibility of function, thus there is an inability to perform any type of operation. Consequently, there is a strong interaction between the performance of lifelines and the building facilities. Evidently, it is a complex problem with various interdependences referring to building and lifeline endurance [39]. Thus, robustness (considering the five main aspects: engineering, organization, social, and economic) requires a multidisciplinary approach that is not strictly related only to engineers, but also to politicians, other public jurisdictions, and social organizations. One can remark that, due to the different interests of stakeholders and the different level of knowledge between them, it is difficult to reach a balance; this requires much time, education, and consent.

Mainly, all research efforts approach the subject of seismic resiliency from a probabilistic way of thinking [36,40,41]. It is a global point of view that is useful to create a policy, offering data and tools for decision making. However, it is also necessary to have a local point of view, specifically to develop, promote, and implement low-damage designs, detailing, and construction technologies, for both new and existing structures [42]. Such a concept is already under way, especially for steel structures [43–45], and other systems [46,47], and was mainly applied after the New Zealand earthquakes. In practice, this second approach is the way to materialize ERnZEBs. Overall, construction technologies associated with structural systems with replaceable fuses, with self-restoring capacity, rocking frames, buckling restrained braces, base isolation, and passive or active damping could ensure an earthquake resilient facility. In addition, the non-structural elements should be independent of the lateral load-resisting system and attached elastically to the main structural system, eventually with sliding capacity within accepted limits of displacement, as well as with elements capable of easy disassembly.

### 3.2. Redundancy

Redundancy is a property of a system composed of more than one subsystem of action, acting as a backup and simultaneously having the ability to have alternative discharge paths. When a subsystem fails, the second subsystem takes action. This concept should be applied to the conformation of the lateral load-resisting system. Another characteristic is that the failure of one subsystem does not cause damage to the other subsystems. This second idea should be used for the detailing of non-structural elements; namely, non-structural elements that do not depend on the behavior of the bearing structure. Overall, the redundancy depicts the capacity of the system to function after the failure of some components in a safe mode (safe-fail).

An ERnZeB should be designed with a lateral load-resisting system, having two "*lines of defense*". For new buildings, related to steel structures as an example, when we construct structural systems such as a steel plate or composite shear walls, in the case of failure of the infill wall (it is the first action of defense against seismic actions), then after that the boundary elements (that shape a steel frame) take all the seismic actions (it is the second action of defense) [48]; see Figure 4. Dual systems, in which they use frames with buckling restrained braces or shear walls, are systems with restoring capacity [49,50]. Related to reinforced concrete buildings, the simplest is the dual frame-shear wall system. Certainly for both steel and reinforced concrete structures, base isolation with supplemental damping provides the most resilient choice for a building [51].

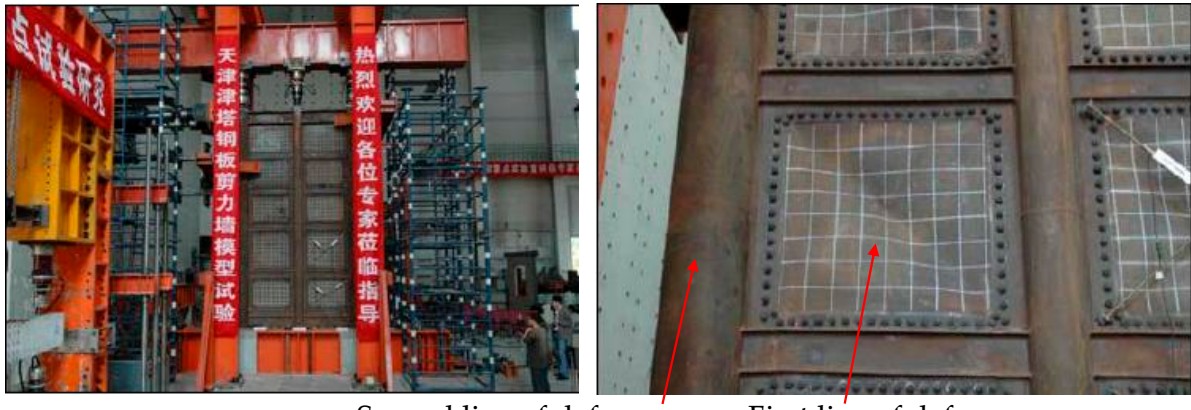

Second line of defense:
*Steel frame*

First line of defense:
*Steel plate*

**Figure 4.** A safe-fail system: steel plate shear wall structural system defined by two lines of defense against severe seismic actions [48].

For existing buildings, a solution to strengthening the structure, combining ease of application without disturbing the users, the ease of replacing the new structure (in case of damage), as well as the development of relative two lines of defense to withstand the seismic actions, is to attach, from the exterior, a structural system that supports the existing one [52]. Towards this direction, exoskeleton structures, see Figure 5, combined, concurrently, with energy-efficient systems (which contribute to the earthquake resistance, i.e., external thermal insulation composite systems, ETICS, see Figure 6, with or without tensile reinforced mortar, TRM, applied to infill masonry) provide solutions that respect the sustainability and resiliency principles [53–57].

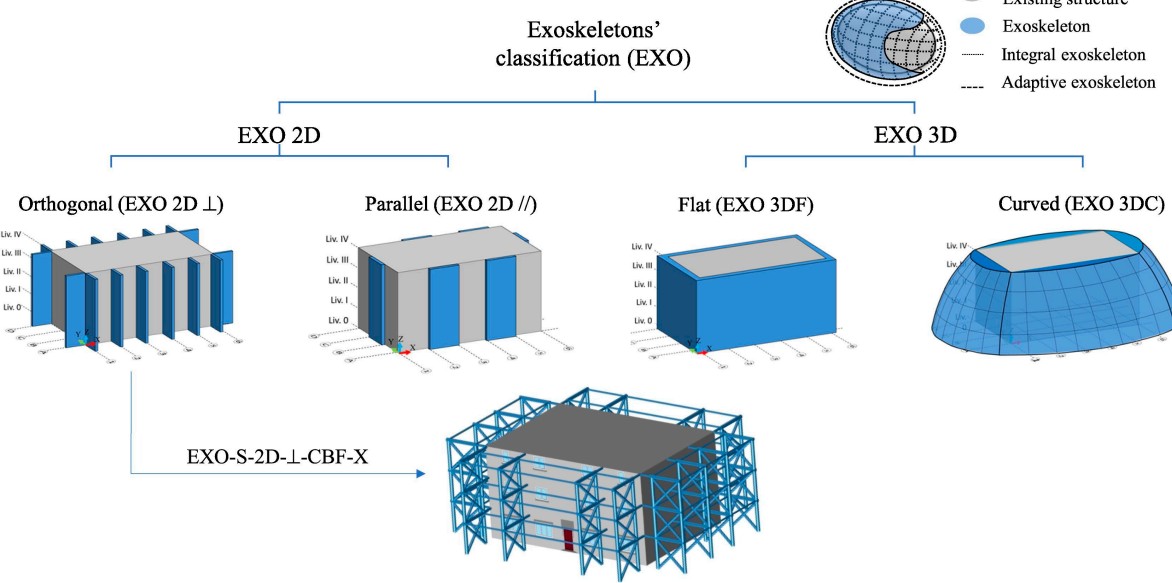

**Figure 5.** Exoskeleton solutions retrofitting building structures [53].

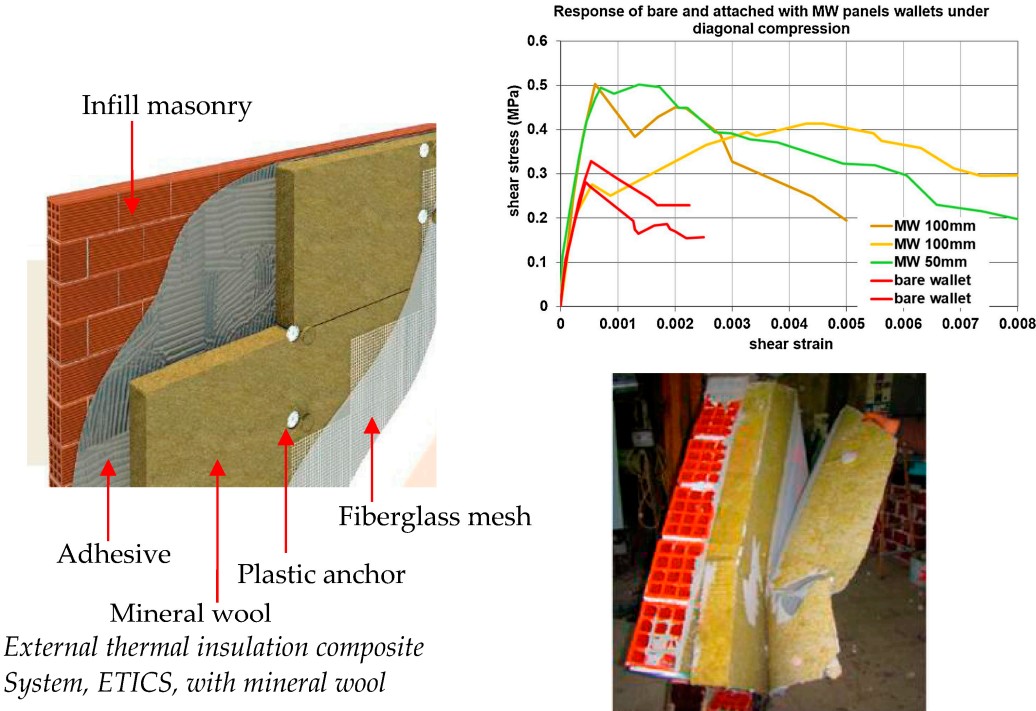

**Figure 6.** External thermal insulation composite system with mineral wool, MW, contributing to the seismic and energy rehabilitation of new and existing infill masonry [57].

*3.3. Resourcefulness*

A resourceful resilient approach has equally the ability to recognize the vulnerable points and turn them into resilient points, through an efficient decision-making system, to manage the priorities in complex situations, and to be prepared for action as well. Certainly, this attribute has a wider meaning, beyond the structural level and going to the community and society levels. For ERnZEBs, this means that a building facility would have the following items: (i) a monitoring system, to detect the behavior, the defects, and the damage; (ii) a plan for maintenance in order to preserve the building in a sound condition; and (iii) a plan for evacuation, through earthquake-resistant escape corridors and compartments.

ERnZEBs, especially for important and sensitive buildings like administrative facilities, fire stations, hospitals, schools, etc., are strongly recommended to have structural and energy monitoring systems, as well as early warning systems. The first one provides us with noteworthy technical information for decision making (diagnosis and prognosis), and the second one provides us with a way of notifying and alarming the building's users (messaging). Such systems are composed of accelerometers, tiltmeters, networking sensors (load cells, strain gauges, transducers, thermistors, thermocouples), properly placed at different building levels, and also of a data acquisition system, communication system, data processing system, data storage system, and cameras as well; see Figure 7 [58,59].

In a building facility's management plan, maintenance is the basic component, targeted to ensure the functionality, safety, and longevity of the building. Already at the design stage, the maintenance plan should be performed and provided with benchmark dates for inspection (i.e., periodic inspections, detailed periodic inspections). The general action plan interacts with structural health monitoring, where the second one diffuses information, thus sometimes alerting for unexpected damage. In this direction, urgent measures could save lives and properties (i.e., the case of the Champlain Towers South collapse on 24 June 2021, in Miami, USA, where a 12 story building suddenly collapsed, causing 98 deaths, with 11 injured and 35 rescued people [60]).

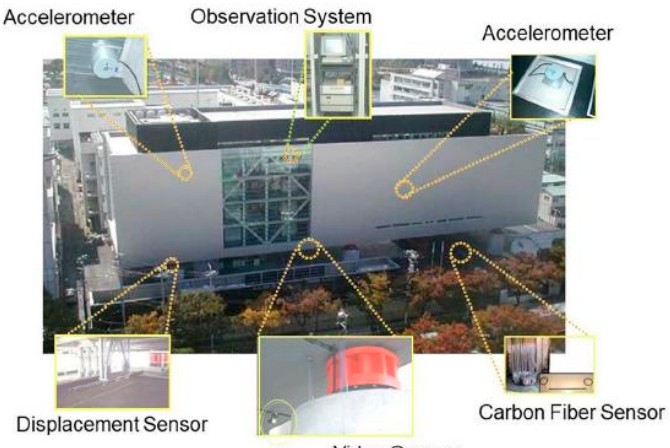

**Figure 7.** Typical example of structural health monitoring for an isolated building [58,59].

ERnZEBs must be structurally conformed, having a stiff reinforced concrete core, within which exist stairs and elevators. It is very important to develop safe pathways for the building's evacuation. Obviously, it is equally important that this escape route remain intact during and until the end of the strong seismic action. Only with very rigid reinforced structural walls forming closed cores can we obtain such a prerequisite. Nowadays, a novel core under the name "SpeedCore", (Coupled Composite Plate Shear Wall-Concrete Filled) [61], concentrates the attributes of a sustainable and resilient system; see Figure 8. The aforementioned is composed of two steel plates, held in position by cross tie bars, filled with concrete, and shear studs ensuring the composite action. Such systems are already used in the steel industry; however, with small adjustments, they could also be used for reinforced concrete structures. Finally, an organizational issue is to post an escape plan at the entrance of the building that can inform the occupants of the route they can follow when the earthquake is over.

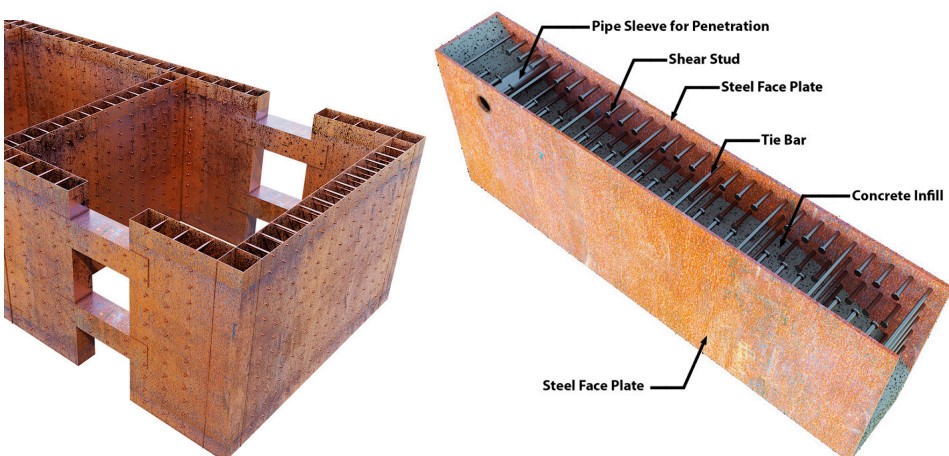

**Figure 8.** "SpeedCore" (Coupled Composite Plate Shear Wall-Concrete Filled) [61].

### 3.4. Rapidity

According to Bruneau et al. [36,37], rapidity refers to "the capacity to meet priorities and achieve goals in a timely manner in order to contain losses, recover functionality and avoid future disruption" or in other words "to the rate at which the community attains its pre-event functionality level". Through this attribute, the time dimension of resiliency is introduced; emphasis is given to the recovery time. This aspect is strongly dependent on society's level of preparedness. A resilient society is made up of resilient lifelines, resilient buildings, and policies that promote and ensure resilience. Focused on buildings,

the main target is to minimize the damage and to maximize solutions that permit rapid structural rehabilitation.

A building facility is an entity of a larger set of buildings that composes a component of society's resilience; thus, each building must be earthquake resilient; this means that it must be designed and constructed implementing such technologies as will offer rapid rehabilitation after a severe earthquake. It is well known that a structure has three main parts: the soil foundation (where the structure is based), the substructure (foundation system), and the superstructure (which is arranged by the lateral load-resisting system and the non-structural elements).

Typically, after a severe earthquake, if the foundation system is aggravated, it requires a lot of time to restore [62–66]. In many cases, demolition (in other respects, of a lightly damaged building) occurs because of the damage, tilting, or sliding of the foundation [67]. Moreover, in many cases, liquefaction problems require much time for soil foundation improvement. Generally, foundation damage is irreversible [68]. Consequently, for an ERnZEB building it is of paramount importance that it is founded on a stable soil (this requires a thorough geotechnical investigation study primarily for the soil characterization and, after that, for the calculation of the bearing capacity not from the soil's capacity but from its deformation capacity); moreover, in cases of near field earthquakes, especially within a radius of approximately 10 Km from the epicenter, as well as in the case of nearby rivers, lakes, sea, the foundation system must be rigid, continuous on the soil foundation, and of box type, with circumferential rigid reinforced concrete walls [62,63,69,70] (i.e., piled raft foundations, piled cap foundations connected with rigid grade beams forming grids, box-type basements, on approximately 3 m depth, with raft foundation, cellular rafts filled with sand). Alternative solutions to protect the foundation system would be the construction of a soil bentonite walls in near proximity to the building [71,72], the execution of vertical trenches of extruded polystyrene sheets [73], or a design of the foundations according to the rocking isolation concept [74–76].

Related to the load-resisting system, we have two options in order to reduce the time to recovery after an earthquake. The first one, as discussed earlier, is to design in a manner that promotes the stiffness, while at the same time following and respecting the rules for a ductile detailing, thus ensuring light damage through rigidity, and collapse prevention through ductile detailing. Dual systems (i.e., reinforced concrete frames and structural walls or steel frames combined with braced frames in both directions, positioned in proper locations, receiving more than 60% of the base shear) will minimize the deformation, and as such will avoid severe or medium damage. Also, base isolation and passive damping are in this direction. The second option is to use replaceable structural fuses where the damage will be concentrated, in those elements, and after a severe seismic action a rapid restoration, through replacement, should be performed; see Figure 9 [43,64,65]. We could simulate the automobile industry, where, when a machine element fails, it is replaced by spare parts that exist in stock. In this way, there will always be a reserve, and after a strong earthquake, critical buildings elements (or structural fuses) will not be constructed, but they will be able to be replaced immediately. Therefore, the recovery of the structural system will be performed within a few days.

With regard to non-structural elements, even if we ensure light damage or even avoid structural damage to the lateral load-resisting frame, we will not avoid damage to the non-structural elements. Therefore, the crucial point is to defend the non-structural elements. In order for a building facility to remain fully operational, it is imperative that the exterior and partition walls, suspended ceilings, ducts, pipes, electrical services, etc. be protected against damage or have the ability to be rapidly restored. The main concept to avoid failure of the non-structural elements is that they be decoupled from the lateral load-resisting system, through the application of gaps between them [77,78] or of use of sliding mechanisms providing relative movement [79–81]. Otherwise, suitable restrainers, through fixing or/and bracing, should be introduced.

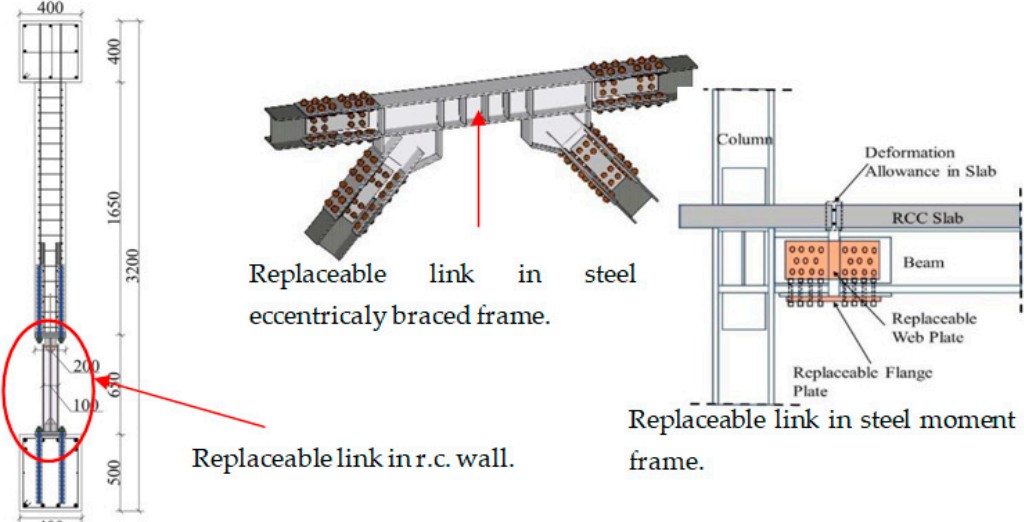

**Figure 9.** Examples of replaceable links [43,64,65].

In this direction, related to the infill masonry walls, the use of external thermal composite insulation systems, ETICS (applying rigid insulation boards such as mineral wool, expanded polyurethane, EPS, or extruded polyurethane, XPS, attached to infill masonry by chemical bonding and mechanical anchoring), are viable solutions providing seismic resilience and energy consumption reduction towards a near zero energy building [57,82]. Moreover, to strengthen the aforementioned system, textile reinforced masonry, TRM, should be used after the placement of the rigid insulation boards [83–88]. Concerning suspended ceilings, proper bracing and hanger systems must be used to avoid collapse [89–93].

Finally, in a holistic view of an ERnZEB building, one can also define the seismic resiliency and proper measurers of the interior content, namely, the furniture and the equipment [94–96]. The furniture will have been held in position, after an earthquake, as well as the equipment, through suitable bracing and anchoring. Moreover, the furniture (i.e., tables) could provide a place where the users could stay safe during the seismic ground motion [97,98].

For example, according to [65], which prescribes a methodology for rating earthquake resilience for buildings, platinum seismic resiliency could be achieved by reaching the following criteria: functional recovery of not more than 72 h, expected economic loss of not more than 2.5% of the building's value, and life safety for occupants, with injures being unlikely. The reader can find in [99–102] practical applications of the methodology provided in [65].

## 4. The near Zero Energy Attribute of an Earthquake Resilient Building

An earthquake resilient building must, at the same time, be a near zero energy facility. This second attribute is related to the energy consumption [103], and further on, a low energy consumption is associated with two types of components, namely: [i] the external thermal insulation composite system and/or external façades that confer, as a function of their composition, earthquake resilience, and [ii] heat and ventilated air conditioning systems, HVAC. The second one is associated with the field of mechanical engineering and is beyond the scope and specialization of the current paper. In any case, the HVAC systems in an earthquake resilient building must be adequately restrained and anchored [104,105].

The main target, respecting the 3 and 4Rs of sustainability and resiliency, is to use systems that have a twofold purpose: improving seismic resilience and reducing energy consumption. Thus, ETICS represents the optimum solution in order to fulfill the above-mentioned conditions. A schematic representation is presented in Figure 10 [106–108]. Studies regarding the seismic performance of such systems are performed in [57,77,78,84–87], which nowadays are mature solutions for commercialization. Moreover, for low-rise build-

ings (of one or two stories), the application of rigid boards of extruded polystyrene would play the role of passive seismic isolation Figure 11 [109]. The lack of a legal framework (i.e., the publication of a European Technical Approval connected with seismic codification and standardization of these systems) is a real obstacle. Currently, ETICS are used only for the insulation of buildings, ensuring near zero energy consumption conditions through the isolation of thermal bridges (details of Figure 10) and the exterior shell as well [110,111].

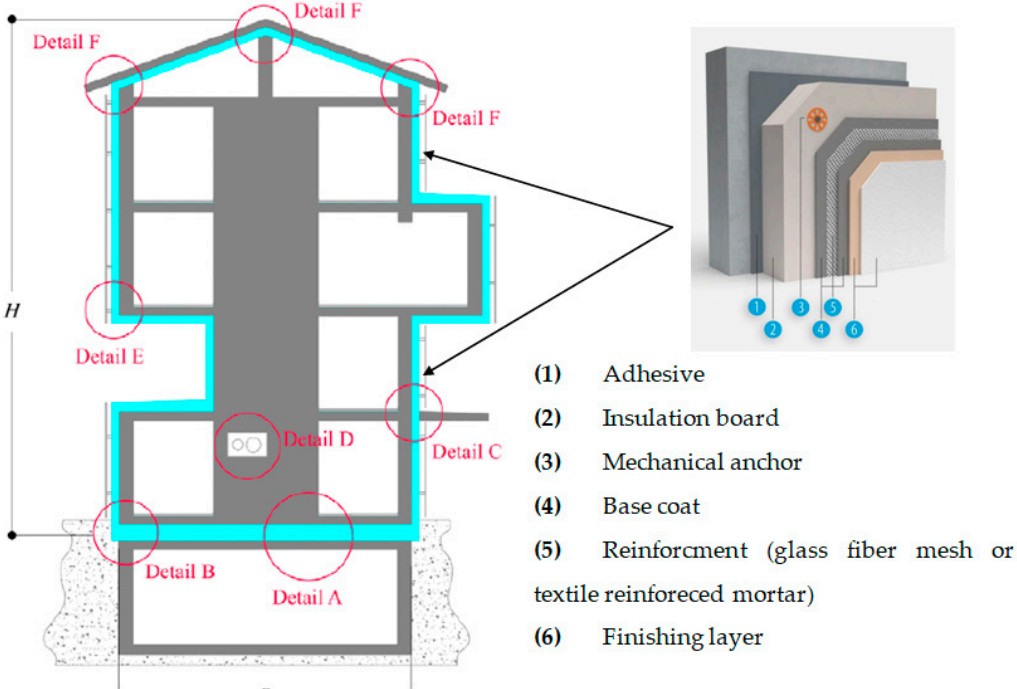

(1) Adhesive
(2) Insulation board
(3) Mechanical anchor
(4) Base coat
(5) Reinforcment (glass fiber mesh or textile reinforeced mortar)
(6) Finishing layer

**Figure 10.** External thermal insulation composite system with dual function, ensuring earthquake resilience through the improvement of the frame's stiffness and the reduction of energy consumption [106–108].

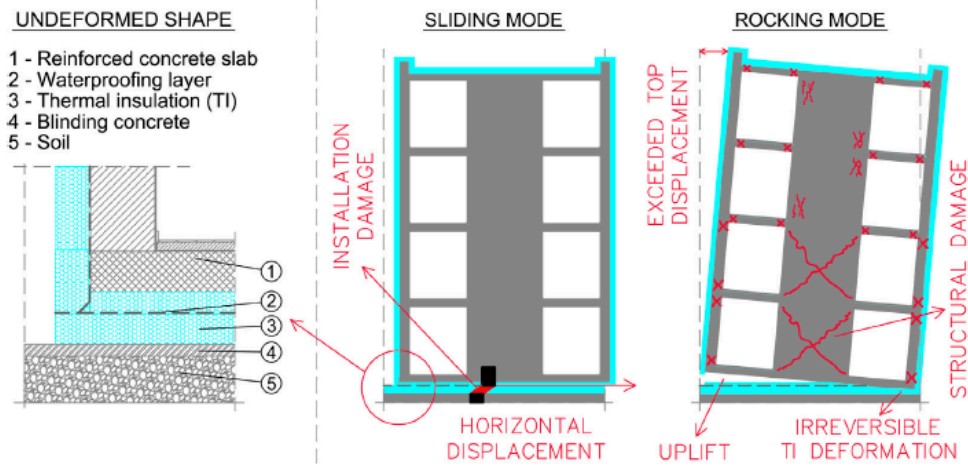

**Figure 11.** External thermal insulation, at the base, acting as a seismic isolation element for one or two story buildings [109].

## 5. Conclusions and Discussion

Learning from earthquakes spanning more than 100 years, one can distinguish the following millstones: we started with an earthquake force-based design (mainly after the San Francisco earthquake, USA, 1906), we passed to earthquake-performance-based design (mainly after the Northridge earthquake, USA, 1994), and nowadays we are developing

earthquake resilient design (mainly after the series of New Zealand earthquakes, 2010). Sustainable (near zero energy) buildings are dictated by the climate crisis due to the fact that the building industry is energy consuming. Sustainability and resiliency are not only connected with building facilities; they form a wider philosophy of doing things that embraces societies and communities and the entire build environment, from the manufacturing of building materials, to urban planning, to the construction of infrastructure and buildings. In order to practically define and apply a sustainable and resilient way of thinking, environmental, social, and economic issues are interrelated in a multidisciplinary manner.

Earthquake sustainability is mainly focused on material consumption, while resiliency is concentrated on the ability of a structural and non-structural system to absorb, recover, and adapt after a destructive seismic action. Both of them strive for society's protection and human wellbeing. A basic component of the built environment is the building facilities. The aforementioned framework creates the need for Earthquake Resilient near Zero Energy Buildings (ERnZEB) conceptual definition. A localized view on practical construction practices, and not only a global probabilistic view, is absolutely necessary. The practical materialization of ERnZEBs requires a proper foundation, structural and non-structural conformation (already performed from preliminary design), low-damage structural systems, durability, and flexible connections with the other architectural finishes and HVAC systems as well. Following the triangle of research, practical application and policies (urban planning, code development, preparedness plan), we will turn the system's vulnerability to resilience and the system's survivability to sustainability.

**Author Contributions:** Conceptualization, A.A. and M.M.; methodology, A.A. and M.M.; writing—original draft preparation, A.A. and M.M.; writing—review and editing, A.A. and M.M.; visualization, A.A. and M.M. All authors have read and agreed to the published version of the manuscript.

**Funding:** This research received no external funding.

**Acknowledgments:** This paper is dedicated to the memory of Victor Gioncu, commemorating the 10th anniversary from his death. A great influencer and a personality who had a profound impact on our engineering thinking and ways to address real life.

**Conflicts of Interest:** Author AA is the owner and manager of the ASAnastasiadis & Associates. The author MM declares that the research was conducted in the absence of any commercial or financial relationships that could be construed as a potential conflict of interest.

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
