# Peer review of "Earthquake Resilient near Zero Energy Buildings: Attributes and Perspectives"

_sustainability, doi:10.3390/su16062317_

Round 1

Reviewer 1 Report

Comments and Suggestions for Authors

The manuscript provides an important review regarding near-zero energy buildings. The manuscript was clean and proper. The author may want to consider the following suggestions.

Line 34: It should be "Major studies were xxxx", please consider it.

Line 53: According to my limited knowledge, soil liquefaction during an earthquake is harmful to superstructures. Whether there is some specific point for near-zero energy building. The authors can consider the following references: "Dynamic responses of a central clay core dam under two-component seismic loading".

Line 385: The layout and caption should be more seriously refined.

Comments on the Quality of English Language

The manuscript was written in good English although some typos and errors still exist. Thus, a minor English polishing is recommended.

Reviewer 2 Report

Comments and Suggestions for Authors

This paper is an interesting review on earthquake resilient near zero energy buildings.  This subject presents novelty.  The text is generally well written and formatted.  Some improvements are suggested, as follows. 

- The literature references seem generally well scientifically established.  However, more references (even 20-30 more references) should be added to demonstrate a high-quality review for the standards of this journal.  For example, more references could be added regarding the effect of soil deformability on the seismic structural response.  

- In the Introduction, which traditionally guides the reader, no references are given in two large paragraphs (lines 72-98).  Please consider improving these paragraphs. 

- This paper mentions "buildings" in general, not discriminating on the basic structural material.  Obviously, reinforced concrete buildings behave differently than masonry, or steel ones.  Also, please give some limits on the size/dimensions of the mentioned buildings.  

- In lines 266, 307 please add reference numbers. 

- In line 279, what is EERI ?  (Please check that all abbreviations should be explained when first used)

- This paper mentions several construction types without organizing them in a rather "confused" way.  Please improve the organization of various building forms.

- The Section 3 seems to provide important information of this article.  However, emphasis is given in the "earthquake" resiliency of buildings, while the "near zero energy" characteristic is not explained adequately.  This review should present a balance between "earthquake" and "near zero energy", while the first one is underlined and the second is not presented obviously.  Please, add more information on "near zero enegy buildings", which serve to improve the novelty of this work. 

- Also, please improve the organization of this work, to help the reader understand the various building forms, characteristics and the dual target of this work. 

Comments on the Quality of English Language

The English Language is generally good.  

Reviewer 3 Report

Comments and Suggestions for Authors

Review of manuscript 2884351:

 Earthquake Resilient Near Zero Energy Buildings: Attributes and Perspectives

 The topic of the manuscript generally fits the purpose of the journal 'Sustainability', although in my opinion it is more suitable for publication in the journal 'Buildings'.

The results of the Authors’ work may be of interest to urban planning teams, designers, contractors and building occupants, in addition to insurance companies. I find no methodological errors in the manuscript.

The solutions presented by the Authors are well illustrated graphically.

The sizes of photos should be standardized.

I recommend for publication after necessary editorial corrections.

Round 2

Reviewer 2 Report

Comments and Suggestions for Authors

The authors have improved this document according to the instructions of the reviewers, so this can be published. 

Comments on the Quality of English Language

The English language is good, just the ussual check before publishing.